# Dietary Exposure and Risk Assessment of Aflatoxin M1 for Children Aged 1 to 9 Years Old in Serbia

**DOI:** 10.3390/nu13124450

**Published:** 2021-12-13

**Authors:** Dragan R. Milićević, Jelena Milešević, Mirjana Gurinović, Saša Janković, Jasna Đinović-Stojanović, Milica Zeković, Maria Glibetić

**Affiliations:** 1Institute of Meat Hygiene and Technology, Kaćanskog 13, 11040 Belgrade, Serbia; sasa.jankovic@inmes.rs (S.J.); jasna.djinovic@inmes.rs (J.Đ.-S.); 2Centre of Research Excellence in Nutrition and Metabolism, Institute for Medical Research, National Institute of Republic of Serbia, University of Belgrade, Bulevar Oslobođenja 18, 11000 Belgrade, Serbia; jelena.milesevic@gmail.com (J.M.); mirjana.gurinovic@gmail.com (M.G.); zekovicmilica@gmail.com (M.Z.); mglibetic@gmail.com (M.G.)

**Keywords:** aflatoxin M1, milk, dairy products, risk assessment, children

## Abstract

The present study was conducted to estimate the exposure and characterize the risk for the child population of Serbia to Aflatoxin M1 (AFM1) from milk and milk-based food. A total of 3404 samples comprising milk and different milk-based food samples were collected from various regions of Serbia from 2017 to 2019. Evaluation of AFM1 exposure was carried out using the deterministic method, whereas risk characterization was evaluated using the margin of exposure (MOE) and the risk of hepatocellular carcinoma (HCC). Detection rates for AFM1 in milk and milk-based food samples ranged between 2% and 79%, with the highest incidence (79%) and mean level (22.34 ± 0.018 ng kg^−1^) of AFM1 being detected in pasteurized and UHT milk. According to the three consumption estimates, the values of estimated daily intake (EDI) were higher for toddlers as compared with children aged 3–9 years. Children aged 1–3 years had the highest risk of exposure to AFM1 in milk, with an estimated daily intake of 0.164 and 0.193 ng kg^−1^ bw day^−1^ using lower bound (LB) and upper bound (UB) exposure scenarios, respectively. Such difference could result from the higher consumption to weight in younger children. Based on the estimated daily intake (EDI) found in this study, the risk of AFM1 exposure due to consumption of milk and milk-based food was low since the MOE values obtained were >10,000. In addition, the risk of HCC cases/year/10^5^ individuals of different age groups showed that the value of HCC, using potency estimates of 0.0017 (mean), was maximum (0.00034) in the age group 1–3 years, which indicates no health risk for the evaluated groups. The present study revealed the importance of controlling and preventing AFM1 contamination in milk through continuous monitoring and regular inspection to reduce the risk of AFM1 exposure, especially in children.

## 1. Introduction

Mycotoxins are toxic compounds produced as secondary metabolites by certain groups of fungi and constitute a significant hazard to food safety and public health [1]. Under certain environmental conditions (i.e., temperature and humidity) and/or biotic stress, toxigenic fungi and their metabolites may contaminate crops and food commodities in different phases of production and processing [2]. Mycotoxins show stability against heat processes, which makes their occurrence in processed food likely expected even if toxin-producing molds are eliminated during the food preparation process [3]. Consumption of mycotoxin-contaminated food may lead to different health adverse effects, including immune suppression, target organ toxicity, genotoxicity, or carcinogenicity [4]. Moreover, when animals ingest these toxins, their metabolites or unmetabolized compounds may be transferred to products such as milk and further contaminate dairy products. Aflatoxin M1 (AFM1) is a principal hydroxylated metabolite of Aflatoxin B1 (AFB1), which may be found in milk from lactating animals after ingesting feed contaminated with AFB1 [5]. The ubiquitous occurrence of mycotoxins in the food chain has been shown in numerous reports over the last decades [6,7]. Thus, to ensure consumer safety due to exposure through food, strict regulations and guidelines have been set by different organizations such as the World Health Organization (WHO) and the European Food Safety Authority (EFSA) to control, measure, and diminish occurrence of the major mycotoxins [8].

Long-term exposure to mycotoxins is associated with myriad health consequences that belong to non-communicable diseases (NCDs), among which are liver and renal cancers, chronic gastritis, and nervous system disorders [9]. Among mycotoxins, aflatoxins represent the major public health concern because they are hepatotoxic, teratogenic, and immunosuppressive. These secondary metabolites are produced by some *Aspergillus* species, especially *A. flavus, A. nomius,* and *A. parasiticus* [10]. The International Agency for Research on Cancer (IARC) classified aflatoxins as Class 1 carcinogenic compounds to humans [11]. According to the risks associated with mycotoxins, the European Union has established the strictest maximum levels for AFM1 (0.05 μg kg^−1^) in raw milk, heat-treated milk and milk for the manufacture of milk-based products and 0.025 μg kg^−1^ in infant formulae and follow-on formulae, including infant milk and follow-on milk [12]. Previous regulations have not eradicated milk AFM1 successfully, which resulted in periodic changes to the official regulations. In the meanwhile, Serbia has set standards for aflatoxins where the maximum regulatory level for AFM1 in raw milk, heat-treated milk, and milk for the manufacture of milk-based products is 0.25 μg kg^−1^ [13]. Dairy products are not included in the Serbian regulation, while for infant formulae and follow-on formulae, including infant milk and follow-on milk, as well as for dietary foods for special medical purposes intended specifically for infants, the permitted level of AFM1 has been set at 0.025 μg kg^−1^.

An important health effect of aflatoxins is their link with liver cancer. In 2012, about 745,000 deaths worldwide were estimated to have been caused mostly by aflatoxin-induced hepatocellular carcinoma (HCC) [14]. In the same year, a total of 782,451 new liver cancer cases and 745,533 related deaths were estimated to occur per year [15]. In addition to liver cancer and cirrhosis, aflatoxins have also been linked to growth stunting in children, malnutrition, kwashiorkor, or marasmus diseases, and the suppression of immune responses [16].

Although, the levels of mycotoxins found in the diet are often low, because of their longer life duration (from now) than adults, children are critically affected by natural contaminants such as mycotoxins and thus are prone to develop chronic syndromes in the future (e.g., mycotoxin-related cancers) [17,18]. Moreover, infants and young children are more vulnerable to the deleterious effects of mycotoxins, because of their larger intake/body weight ratio, higher metabolic rate, and lower detoxification capabilities [19,20]. Therefore, it is necessary to evaluate mycotoxin presence in foods and the level of exposure in children [21].

Food security and food safety is an important prerequisite for good health. Milk and dairy products are a source of many nutrients including proteins, fatty acids, calcium, vitamins, and minerals essential for human health, especially in infants and children [22]. However, the risk of contamination by AFM1 is an important food safety concern for milk. Despite the available data on AFM1 occurrence in milk and dairy products, information on exposure and risk assessment of infants and young children in Serbia is lacking. This is due to a combination of limited monitoring systems and a lack of food consumption data. Thus, the extent and health implications associated with mycotoxin exposure of infants and young children need to be evaluated and should be given a priority in Serbia.

Therefore, the objective of the present study was to conduct a preliminary risk assessment and evaluate the dietary exposure of the child population in Serbia to AFM1. The results of our study are helpful to risk managers in their prioritization for food monitoring programs as part of risk-based food control, as well as in the application of adequate measures to protect the health of children.

## 2. Materials and Methods

### 2.1. Sample Collection

We conducted a risk assessment for AFM1 by combining the concentration of AFM1 in food commodities from several studies published between 2017 and 2019 [5,23,24]. In brief, a total of 3404 milk and milk-based food samples was randomly collected from various regions of Serbia from 2017 to 2019. The samples consisted of different types of milk, dairy products, and infant formula. The majority of collected samples were from local dairy processing plants that manufacture fluid milk, cheese, cream, or cultured dairy products, while some of the food samples (i.e., infant formula, milk beverages) were from imported sources.

### 2.2. Sample Preparation and Analysis

The sample analysis was performed with an enzyme-linked immunosorbent assay (ELISA). Preparation of the samples and the ELISA test procedure for the determination of AFM1 in milk-based food samples was performed according to the manufacturer’s instructions (Tecna S.r.l., Mirandola (MO), Italy). For ELISA analysis, 100 µL of diluted antibody solution was added to each well and shaken for 30 s. The plate was incubated for 45 min at room temperature (20 to 25 °C). After four washing steps, 100 µL of enzyme conjugate solution was pipetted into separate duplicate wells and the plate gently shaken to mix. After incubation of the plate for 15 min at room temperature in the dark, the liquid in the wells was discarded, and, to complete removal of the remainder of the liquid, the plate was tapped against an absorbent paper (three times). After four washing steps, 100 µL of developing solution was added to each well and the plate was shaken for 30 s. The plate was incubated for 15 min at room temperature in the dark. Then, using a multichannel pipette, 50 µL of the stop solution reagent was added to each well and mixed for several seconds.

Optical density was measured using ELISA-reader Thermo Scientific (Waltham, MA, USA, SAD) model 364, at a wavelength of 450 nm. Ascent software (v.1.0) was used for data acquisition and processing. The detection limit of the method was 0.005 μg/kg, while specificity was 100%, and 16% for AFM1 and AFM2, respectively. Relative standard deviation of reproducibility was 6%, and recovery was 110%. Quality assurance regarding the ELISA method was confirmed by participation in a proficiency testing scheme (PROGETTO TRIESTE) of lyophilized milk. The proficiency test results were satisfactory according to the calculated z-score of 0.09 and 1.27 for AFM1 and AFM2, respectively (acceptable range for z: −2 to 2).

The samples with AFM1 levels above the MRL were confirmed and also quantified by LC-MS/MS analytical techniques.

### 2.3. Extraction of Milk Samples for LC-MS/MS

LC-MS/MS analysis of AFM1 was carried out according to the method previously published by Milicevic et al. [5,23,24].

#### 2.3.1. Standard Solution Preparation

AFM1 standard was purchased from Sigma Aldrich Chemical Company (St. Louis, MO, USA). Working solutions, prepared by diluting the stock solution, were used to prepare the calibration curve and to spike milk samples. The final concentrations of AFM1 used in the calibration curve were 0.2, 1.0, 2.5, 10.0 and 20.0 ng/mL. All stock and working standard solutions were stored in brown vials at −18 °C. For recovery studies, defatted milk was enriched with AFM1 working standard solution at three spiked levels: 0.025, 0.05, 0.075 μg/kg (i.e., 0.5 times MRL, MRL and 1.5 times MRL).

#### 2.3.2. Chromatographic and MS Parameters

The instrument used for LC-MS/MS was a Waters Acquity UPLC system (Waters, Milford, MA, USA) coupled with a TQD mass spectrometer (Waters Micromass, Manchester, UK). A Purospher Star (Merck, Darmstadt, Germany) RP-18 column (50 × 2.1 mm, 2 μm particle size) was used for the separation of AFB1. The mobile phase was 0.1% acetic acid and methanol (35:65). Isocratic flow was maintained at 0.3 mL/min. Two product ions were monitored (329 > 273 Da and 329 > 259.1 Da). Quantification ion was 273 Da. MassLynx 4.1 software was employed for data acquisition and processing. The detection limit of the method was 0.02 μg/kg, relative standard deviation of reproducibility was 5.4%, and recovery was 65–81%. Linear regression analysis was performed using JMP v.10 software.

### 2.4. National Food Consumption Survey on Toddlers and Children

A Serbian National Food Consumption Survey on toddlers and children was conducted between 2017 and 2021 according to the EFSA EU MENU methodology [25]. Valid data were collected from a total of 576 participants with 290 toddlers aged from one to below three years old and 286 children aged from three to nine years old. Data collection was conducted using project-specific national survey pack that included a general questionnaire, an age-appropriate food propensity questionnaire (FPQ), and a 24 h food diary. The consumed portion sizes were estimated based on natural units, household measures, packaging information and country-specific portion size measurement aid (PMSA) (i.e., previously tested Food Atlas) [26]. Following EFSA guidance on the EU Menu methodology, a previously developed and validated innovative nutritional software tool DIET ASSESS and PLAN (DAP) was used [27] for standardized and harmonized food consumption data collection and comprehensive dietary intake assessment. Basic FoodEx2 codes including implicit facets were assigned to all foods and recipes from the Serbian Food Composition Data Base (FCDB) which is integrated into the DAP platform. Weight measurements were obtained for children without shoes and jackets using a digital balance and data were recorded to the nearest 0.1 kg. For children’s height measurements, portable stadiometers were applied with 0.1 cm accuracy.

### 2.5. Health Risk Assessment

Deterministic methods (or single point) were employed to derive a worst-case risk estimate. Assessment of cumulative risks posed to the health of children by consumption of milk and milk-based food was performed in three stages which comprised exposure assessment, risk characterization, and assessment of liver cancer risk.

### 2.6. Exposure Assessment

Chronic AFM1 exposure among the two age groups was estimated by the deterministic approach involving the average probable daily intake (APDI) method [28]. Exposure was calculated for all the food categories, and for both consumer groups according to their gender and age to highlight the differences in exposure. The EDI of AFM1 (expressed as ng kg^−1^ bw day^−1^) was calculated based on the concentration of AFM1 detected and the intake rate of analyzed foods, according to Equation (1):*EDI* = Σ*c* ∗ C/*bw*(1)
where Σ*c* is the average concentration of AFM1 (ng kg^−1^), C is the daily average consumption of the commodity (kg per day), and *bw* is the body weight for the male and female child populations (kg).

The mean concentrations of AFM1 in selected milk and dairy products were taken from Table 1. Within the general framework of chemical risk assessment, a difficult step in dietary exposure evaluation is handling concentration data reported to be below the limit of detection (LOD). These data are known as non-detects and the resulting occurrence distribution is left-censored. The left-censored data (data below LOD and LOQ) were processed by applying EFSA’s substitution method [29]. According to this guidance, for dietary exposure assessments, three exposure scenarios were considered. Middle bound (MB), assuming that the not detected results correspond to half of the LOD (ND = 2.5 ng kg^−1^) was used for all AFM1 when a finding with a value <LOD was in ≤60% of samples. In contrast, when a large percentage of the results were below the LOD (>60 but ≤80% non-quantified and with at least 25 results quantified), two estimates used a lower bound (LB) scenario, in which zero was assigned to samples showing AFM1 concentration below LOD/LOQ, and the upper bound (UB) was obtained assuming the value for the LOD of AFM1 (5.0 ng kg^−1^) for the results of AFM1 reported as not detected (ND = LOD). Furthermore, following EFSA recommendations, exposure calculations at the 95th percentile (P95) of AFM1 concentration (P95) were performed to evaluate the worst-case scenarios [30]. The daily average consumption of these products and mean body weights were obtained from the data provided in the food frequency questionnaire by age (Table 2 and Table 3). The different food commodities were grouped within each food category to better explain their contribution to the total dietary exposure to AFM1.

### 2.7. Risk Characterization

Since AFM1 is considered carcinogenic, there is no TDI based on a dose of no observable effect (NOEL). Therefore, risk characterization originating from the oral exposure to aflatoxins was calculated using two approaches; the qualitative margin of exposure (MOE) approach established by EFSA [30] for substances that are both genotoxic and carcinogenic and the quantitative approach to liver cancer risk estimation proposed by the FAO/WHO [31]. 

The MOE value was calculated using Equation (2): MOE = BMDL_10_/EDI(2)
where BMDL10 is the benchmark dose lower confidence limit (BMDL10) for 10% increased cancer risk. Based on animal data, EFSA concluded that AFM1 induces liver cancer with a potency one-tenth that of AFB1 (for AFB1 0.4 μg kg^−1^ bw per day^−1^), so hence, a potency factor of 0.1 for the AFM1 risk assessment was used in this study. EDI is the average daily intake used to estimate chronic dietary exposure to AFB1, as calculated in Equation (1). A calculated MOE value lower than 10,000 implies that exposure to a carcinogenic and genotoxic substance contributes to the risk of HCC and is of concern to public health [30]. 

### 2.8. Assessment of Liver Cancer Risk—The Carcinogenic Potency

Most health concerns for aflatoxins are related to primary liver cancer burden, as the ingestion of these toxins has been directly linked to HCC development, particularly in individuals infected with hepatitis virus. To estimate the risk of cancer posed by dietary exposure to AFM1, we used the following equation: Population risk = EDI × Average potency(3)

Regarding the differences in carcinogenic potency, for AFM1, according to JECFA [28], AFM1 induces liver cancer with one-tenth of the potency of AFB1. Therefore, the carcinogenic potency (CP) of AFM1 was calculated to be 0.0562 additional cancer cases per 100,000/year per 1 ng kg^−1^ bw day^−1^ for hepatitis B virus (HBV) surface antigen positive (HBsAg^+^) populations and 0.0049 additional cancer cases per 100,000/year per 1 ng kg^−1^ bw day^−1^ for HBsAg^−^ populations. The prevalence used of HBV-infected individuals in Serbia was 1%, based on earlier studies [5]. Thus, the CP of 1 ng AFM1 kg^−1^ bw day^−1^ in a population with a 1% prevalence of HBV infection would be 0.005413 cases per year per 100,000 people according to Equation (4): Average cancer potency = (0.0049 × 0.99 + 0.0562 × 0.01)(4)

### 2.9. Statistical Analysis

Data were analyzed using Minitab statistical software version 17 (Minitab Ink., Coventry, UK). AFM1 concentrations for the studied samples were expressed in the form of descriptive statistics and presented in Table 1, Table 4, Table 5, Table 6 and Table 7. A Shapiro–Wilk test of normality was run to check the normality of data and after recording the data as normal, a further test was used for statistical evaluation of the data.

## 3. Results

### 3.1. Prevalence of AFM1 in Milk and Milk-Based Food

The prevalence of AFM1 in milk and milk product samples collected from various regions of Serbia from 2017 to 2019 is presented in Table 1. Amongst the collected samples, 574/725 pasteurized and UHT milk, 67/201 milk powder, 158/775 fermented milk products, 145/714 milk beverages, 14/92 infant formula, 19/132 sour cream, 13/90 whey, 14/143 butter, and 7/404 cheese were contaminated with AFM1. Overall, the mean levels (ng kg^−1^) of AFM1 based on the LB mean ranked as follows: pasteurized and UHT milk > whey > fermented milk products > milk powder > sour cream > butter > milk beverages > infant formula > cheese. As expected, the highest incidence of contamination (79%) and the greatest mean concentration of AFM1 were observed in pasteurized and UHT milk (22.34 ± 0.02 ng kg^−1^), while cheese with 1.36 ± 0.01 ng kg^−1^ showed the lowest mean concentration. Among the different milk products, the maximum AFM1 level found in this study was registered in a whey sample, reaching a contamination level of 278 ng kg^−1^, followed by cheese (276 ng kg^−1^) and a fermented milk product (174 ng kg^−1^). 

The mean concentration of AFM1 in the present study is slightly lower compared to the previous studies from Serbia [32,33,34]. In addition, the results of this study are in agreement with the reported AFM1 concentrations in milk and dairy products from global studies, where the prevalence of AFM1 in milk worldwide was 79.1% [30]. This could be explained by the fact that preventive and control activities during harvest, processing, and storage of dairy feeds, combined with the improvement of risk management actions in dairy processing industries have been improved in recent years to a considerable extent. The variation in the mean AFM1 contamination in milk previously reported may be attributed to differentiation in carry-over rates of AFB1 in milk, which depends on the animal species, but these rates can also vary greatly depending upon nutritional, environmental, and physiological factors such as stage of lactation, systemic diseases, local (mammary) infections, level of AFB1 in feed, rate of feed ingestion, and geographical and seasonal conditions [35]. It is also important to highlight that many of the data have been obtained using different methodologies, with a consequence of different sensitivity and precision. 

Aflatoxin contamination of foods of animal origin (milk, dairy products, eggs, and edible animal products) is a global public health and economic concern. The presence of AFM1 in milk and milk products is most probably the consequence of feeding dairy cows a diet contaminated with AFB1. Since their presence has been responsible for significant adverse health and economic issues affecting consumers and farmers worldwide, the formulation of regulations to control their presence in animal feed has been triggered [8]. Various investigations conducted in Serbia in the last decade have revealed a significant presence of aflatoxins in maize [3,36]. In general, the reported concentration of AFB1 in maize and consequently the presence of AFM1 in milk showed year-to-year variations in AFM1 prevalence [24]. Therefore, to estimate the risk of illness in the Serbian population exposed to aflatoxins, a series of survey studies have been conducted to monitor the incidence of AFM1 contamination, particularly in raw milk in Serbia.

Albeit mean concentrations of AFM1 in our study were lower in whey and cheese than in the milk samples, the highest concentrations of AFM1, which we measured in whey and cheese, was also observed in these two products in previous studies, which concluded that during cheese production, 60% of the initial content of AFM1 accumulates in the whey, while 40% of the AFM1 remains in the curd or fresh cheese [23,37,38]. This might be due to the water-soluble nature of AFM1 and its affinity to form a hydrophobic bond with the hydrophobic part of casein that is subsequently concentrated in cheese [39]. Furthermore, the AFM1 concentration in soft cheeses was generally 2.5–3.3 times higher, and in hard cheeses, 3.9–5.8 times higher, than in the milk from which the cheeses were made [34,37,40]. Most studies have reported that AFM1 concentrations in milk products are strongly dependent on the AFM1 concentrations in milk, and hence milk concentrations could be a good predictor of the AFM1 concentration in cheese and whey. Notably, in the present study, AFM1 was found in 20% (158/775) of fermented milk products (ranging from 25 to 174 ng kg^−1^, mean 9.58 ± 0.02 ng kg^−1^). The presence of AFM1 in fermented milk products may be due to manufacturers usually using imported dry milk for producing dairy products that were contaminated with AFM1. However, this low level of AFM1 in fermented milk products could also be attributed to the function of lactic acid bacteria, during fermentation [41]. Results indicated that the incidence and mean AFM1 values obtained in the present study are low to moderate. Hence the risk of AFM1 exposure could not be a public health concern for the general population. However, as children use milk and dairy products in their diets frequently and are more sensitive to the adverse effects of aflatoxins compared to adults, ingestion of low doses of AFM1 in milk over long periods must be considered a risk, and should not be underestimated or neglected.

### 3.2. Dietary Exposure Assessment

Risk assessment through dietary exposure is the process of estimating the magnitude and the probability of a harmful effect on individuals or populations from specified agents or activities. Per definition, exposure assessment, as one component of risk assessment methodology, combines mycotoxin levels in food with consumption patterns, and therefore, provides valuable information for risk management if mycotoxins compromise food safety and health hazards, at either an individual or a population level [9]. Following the recommendations of EFSA [29], the current study utilized the most comprehensive (chronic) exposure scenario to assess the EDI of AFM1 by Serbian children, taking into account a range of LB and UB values.

Based on the data described before (Section 2.4), the EDI of AFM1 (ng kg^−1^ bw day^−1^) through milk and dairy product consumption in different age categories was calculated and is presented in Table 4. It is widely considered that the LB scenario generally underestimates contamination and exposure levels and that the UB scenario overestimates them [29]. As can be seen from Table 4, the exposure of AFM1 differs from product to product, and a significant difference (*p* < 0.05) between the exposure values assessed, considering the UB, and LB scenarios, was found within products. In consequence, for these purposes, the middle-bound approach should be applied. In this study, the highest EDIs of AFM1, i.e., 0.164–0.172 ng kg^−1^ bw day^−1^ (LB-UB) and 0.193–0.202 ng kg^−1^ bw day^−1^, were found for the consumption of pasteurized and UHT milk by male and female toddlers, respectively.

The AFM1 exposures were ranked for all the food types: pasteurized and UHT milk > whey > fermented milk products > milk beverages > sour cream > infant formula > butter > cheese. The food categories pasteurized and UHT milk (46 to 48%) and fermented milk products (27 to 31%) were the main contributors to the overall AFM1 mean exposure throughout both age groups (Figure 1). Due to the limited number of consumption and concentration data for milk powder and clotted cream, these food categories were not taken into account for risk assessment.

The findings obtained in this study showed a remarkably lower exposure of the Serbian child population in comparison with the estimates of AFM1 intake reported by Kos et al. [32] and Milićević et al. [5]. These contradictory results regarding the EDI could be attributed to some uncertainties, including occurrence data (sampling strategy, low number of samples, seasonal effects, lack of sensitivity of some analytical methods) and exposure modeling. In addition, data from a health survey of the population of Serbia indicate a negative trend in the consumption of milk and dairy products. At least 41.8% of the population consumed milk and dairy products on a daily basis in 2021, which is significantly less than in 2013, when 51.7% of the population reported daily consumption. 

Although aflatoxin contamination in food occurs in many countries around the world, the nations that have been identified to be substantially exposed to AFM1 (sometimes dramatically) are primarily in sub- Saharan Africa and South Asia (Iran and Pakistan), and of particular concern are populations of children [42]. Generally, the mean dietary AFM1 exposure from milk and dairy product consumption in European populations is comparatively low, which may be the result of strict regulations on mycotoxins in feed and milk products and from the adoption of an integrated food safety management system. In comparison to international studies, our results were lower than the results of several studies. In addition, the current EDI values did not exceed the previously established international TDI limit (0.2 ng kg^−1^ bw day^−1^) [43]. Although the estimated AFM1 exposure levels for milk and dairy product consumers in the present study are relatively low, owing to the genotoxic and carcinogen nature of aflatoxins, the approach of “as low as reasonably achievable” (ALARA) could be adopted in forthcoming regulations to protect Serbian consumers against the health effects caused by AFM1. 

### 3.3. Risk Characterization/Cancer Risk Attributable to AFM1

The risk of exposure to AFM1 through milk and dairy product consumption was characterized using MoE (Table 5), and the liver cancer risk approach (Table 6 and Table 7). According to the EFSA scientific committee guidance [30], when the MoE value is ≥ 10,000, it is considered that there is a low risk of a negative impact on public health. Our results showed that MoE values for LB and UB exposure scenarios to AFM1 were far higher than 10,000 in toddlers and other children, which indicates no health concern due to exposure to AFM1 through consumption of milk and dairy products. However, as children consume more milk relative to their body weight, children’s exposure risk to AFM1 in milk and dairy products should be a continuous focus of attention.

The results of the characterization of HCC risk (cases per 100,000 individuals per year) for different age groups due to AFM1 exposure based on the calculation of the risk by P-cancer and EDI, are presented in Table 6 and Table 7. The additional cancer risk due to mean exposure to AFM1 associated with milk and dairy product consumption in toddlers using potency estimates of 0.0017 (mean) for the LB scenario ranged from 0.00032 to 0.00001 and from 0.00038 to 0.00001 cases per 100,000 individuals per year for males and females, respectively. For other children, the mean estimated number of liver cancer cases for the LB scenario ranged from 0.00030 to 0.00012 cases per 100,000 individuals per year for males and from 0.00017 to 0.00001 cases per 100,000 individuals per year for females. The main contribution of HCC risk due to AFM1 exposure was caused by the consumption of pasteurized and UHT milk, estimated at 0.00038 and 0.00039 cases per 100,000 individuals per year for the LB and UB scenarios, respectively. Our results are considerably lower than those reported in an assessment by EFSA [30] where the estimated cancer risk (mean and UB) ranged between 0.002–0.035, 0.008–0.032, 0.003–0.018, 0.001–0.006, 0.001–0.004, and 0.001–0.003 aflatoxin-induced cancers per 100,000 person-years for infants, toddlers, other children, adolescents, adults, and the elderly, respectively. Globally the standardized annual incidence rate for liver cancer is 15.3 per 100,000 among men and 5.4 per 100,000 among women [44]. Several studies conducted in African and South Asian countries have investigated the health impacts of early dietary exposure to aflatoxins. Prolonged exposure to aflatoxin might be the underlying cause of congenital disabilities and child growth impairment. Most of the studies have reported that exposure to aflatoxins might be the underlying cause of child growth impairment [45,46]. Nonetheless, the possible association between chronic exposure to aflatoxins in early life and the early onset of hepatic cancer has been explored by several studies. AFB1 is the most potent human hepatocarcinogen, accounting for around 4.6–28.2% of the total HCC cases worldwide [47]. Further, there is a strong synergistic association between AFB1 and HBV infection in the etiology of HCC. Recent results from a national study in Serbia [48] revealed that the rate of acute cases of HBV infection continued to decline (incidence of 1.25/100,000 inhabitants) over the last few years (2010–2019), which is following global trends and most likely reflects the impact of national vaccination programs. On the other hand, there is an increasing trend in the numbers of registered cases of chronic HBV and hepatitis C infections. Improving the health and well-being of children are priority health policies of many countries. It is necessary to provide children with stability and an environment for growth and development that includes good health and proper nutrition. Numerous epidemiological studies link childhood health with health outcomes in adults, and investing in children’s health is one of the most important measures that society can take to improve the health of the entire population. In summary, future work in this area would focus on the survey of occurrence and exposure to AFM1 to identify geographic regions where AFB1 levels in staple food are high enough to cause concern for human populations.

## 4. Strengths and Limitations

This is the first ever conducted study on AFM1 exposure risk assessment of children population through milk and milk products in Serbia using harmonized food consumption data collected within the EU Menu project survey according to EFSA guidance methodology, which makes this data comparable with other harmonized food consumption data in whole Europe. A limitation of this study is that relatively small number of infant formula, clotted cream, butter and sour cream have been considered, leading to underestimated health risk associated with exposure to AFM1 from milk products among the children. Furthermore, limitation of the study lies in the fact that total exposure to AFM1 was not assessed for the whole diet, i.e., from other food groups that certainly contain AFM1, and will be the area of research in further studies.

## 5. Conclusions

Considering the present evidence on the negative health effects of AFM1, this study through the MOE approach and the population risk assessment method suggests that milk and dairy products had negligible health risk to the child population due to AFM1 exposure. Despite current AFM1 concentrations being not high enough to elicit toxic effects, risk data should be interpreted carefully due to the present study investigating only the consumption of milk and dairy products. Thus, the focus of future studies should be on exposure from complete diets commonly consumed by Serbian children to estimate cumulative exposure from all sources of aflatoxins. In addition, further research is advisable, in particular related to the association of liver cancer with AF intake and HBV infection. Since the contamination of feedstuffs with AFB1 plays a major role in the contamination of milk, the government and all stakeholders involved in the milk supply chain should pay more attention to implementing an integrated food safety management system to prevent the production of mycotoxins in dairy cattle feed and to reduce AFM1 residues in milk.

## Figures and Tables

**Figure 1 nutrients-13-04450-f001:**
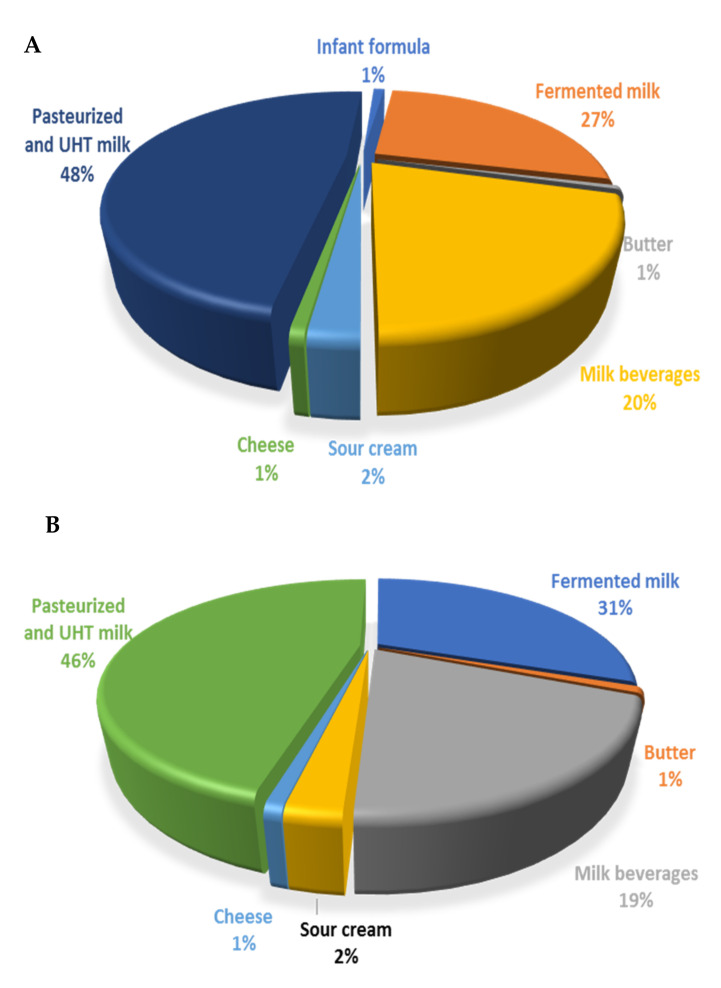
Contribution (%) of the most important food groups to the long-term dietary exposure of children aged 2 to 6 years (**A**) and toddlers, 1–3 years (**B**), to AFM1.

**Table 1 nutrients-13-04450-t001:** Aflatoxin M1 incidence and concentration in milk and dairy product samples included in the study.

Type of Sample	*n*/*N* (%)	Mean (ng kg^−1^ ± SD) of All Samples				Mean Positives (ng kg^−1^ ± SD)	Median Positives (ng kg^−1^)	Q1 Positives (ng kg^−1^)	Q3 Positives (ng kg^−1^)	Range (ng kg^−1^)
		LB	MB	UB	P95					
Infant formula	14/92 (15.2)	1.6 ± 0.004	3.76 ± 0.003	5.88 ± 0.002	12.5	10.0 ± 0.002	11.0	8.00	13.00	8.0–14.0
Fermented milk products	158/775 (20.3)	9.58 ± 0.02	11.5 ± 0.02	13.56 ± 0.02	57.0	47.0 ± 0.022	38.0	34.0	56.25	25.0–174.0
Clotted cream	0/48	-	-	-		-	-	-	-	<5.0
Butter	14/143 (10)	5.20 ± 0.01	7.44 ± 0.016	9.70 ± 0.01	47.0	53.0 ± 0.016	47.0	41.75	58.0	39.0–92.0
Milk beverages	145/714 (20)	4.22 ± 0.01	6.21 ± 0.011	8.20 ± 0.01	23.0	20.77 ± 0.02	14.0	10.50	22.50	5.0–117.0
Sour cream	19/132 (14)	6.90 ± 0.02	9.04 ± 0.018	11.18 ± 0.02	48.0	47.95 ± 0.002	39.0	31.0	60.0	25.0–103.0
Cheese	7/404 (2)	1.36 ± 0.01	3.82 ± 0.014	6.28 ± 0.01	5.0	7.89 ± 0.08	49.0	40.0	58.0	39.0–276.0
Pasteurized and UHT milk	574/725 (79)	22.34 ± 0.02	22.87 ± 0.018	23.40 ± 0.01	53.0	28.22 ± 0.016	25.00	19.00	35.0	5.0–132.0
Milk powder	67/201 (33)	9.12 ± 0.02	10.79 ± 0.020	12.46 ± 0.02	47.0	27.37 ± 0.03	16.00	9.00	36.00	5.0–155.0
Whey liquid	13/90 (14)	14.82 ± 0.05	16.96 ± 0.05	19.10 ± 0.05	70.0	102.6 ± 0.10	70.0	20.50	211.0	5.0–278.0
**Total**	**1012/3404 (29.7)**	**9.47 ± 0.02**	**11.23 ± 0.02**	**12.99 ± 0.02**	**44.0**	**31.86 ± 0.026**	**27.00**	**16.00**	**38.00**	**5.0–278.0**

*N* = number of analyzed samples. *n* = number of positive samples (AFM1 > LOD). % = percentage of positive samples. The limit of detection (LOD) for AFM1 is 5.0 ng kg^−1^. Lower bound (LB) = assuming that the not detected results are equal to 0 (ND = 0). Middle bound (MB) = assuming that the not detected results correspond to half of the LOD (ND = 2.5 ng kg^−1^). Upper bound (UB) = assuming that the not detected results correspond to the LOD (ND = 5.0 ng kg^−1^). P95 = 95th percentile. First quartile (Q1) 25% of the data are less than or equal to this value. Second quartile (Q2) = the median. A total of 50% of the data are less than or equal to this value. Third quartile (Q3) = 75% of the data are less than or equal to this value.

**Table 2 nutrients-13-04450-t002:** Characteristics of the study sample [25].

Age Group	Body Weight (kg)		*N*	
	Male	Female	Male	Female
Toddlers, 1–3 years	14	13	98	91
Children, 3–9 years	24	24	159	150

*N* = number of participants.

**Table 3 nutrients-13-04450-t003:** The average intake of food groups by child population (g/day) [25].

Age Group		Infant Formula	Fermented Milk Products	Clotted Cream	Butter	Milk Beverages	Sour Cream	Cheese	Pasteurized and UHT Milk	Whey Liquid
Toddlers. 1–3 years	M	31.51	133.40	9.94	4.88	230.00	15.00	22.50	102.87	-
	F	29.93	115.76	8.72	5.30	213.30	17.47	24.40	112.05	-
Children. 3–9 years	M	-	153.24	13.31	7.95	220.83	16.58	27.19	99.55	250.0
	F	-	150.50	13.60	8.02	199.75	15.05	26.94	93.58	-
**Average**		**30.72**	**138.23**	**11.41**	**6.54**	**215.97**	**16.02**	**25.25**	**102.01**	**250.0**

M = male. F = female. Food groups were categorized according to a national survey.

**Table 4 nutrients-13-04450-t004:** Estimated daily intake (ng kg^−1^ bw day^−1^) of AFM1 in selected food products for two age categories.

Food Group	Exposure (ng kg^−1^ bw day^−1^)											
	Toddlers, 1–3 Years						Children, 3–9 Years					
	Male			Female			Male			Female		
	LB	UB	P95	LB	UB	P95	LB	UB	P95	LB	UB	P95
Infant formula	0.004	0.014	0.029	0.004	0.014	0.029						
Fermented milk products	0.091	0.129	0.543	0.085	0.121	0.508	0.061	0.087	0.364	0.060	0.085	0.358
Butter	0.002	0.003	0.016	0.002	0.004	0.019	0.002	0.003	0.016	0.002	0.003	0.016
Milk beverages	0.069	0.135	0.378	0.069	0.134	0.377	0.039	0.075	0.212	0.035	0.068	0.191
Sour cream	0.007	0.012	0.051	0.009	0.015	0.065	0.005	0.008	0.033	0.004	0.007	0.030
Cheese	0.002	0.010	0.008	0.003	0.012	0.009	0.002	0.007	0.006	0.001	0.007	0.005
Pasteurized and UHT milk	0.164	0.172	0.389	0.193	0.202	0.457	0.093	0.097	0.220	0.087	0.091	0.207
Whey liquid							0.154	0.177	0.199			
**Total**	**0.340**	**0.475**	**1.415**	**0.365**	**0.501**	**1.463**	**0.201**	**0.277**	**0.850**	**0.190**	**0.262**	**0.807**

Lower bound (LB) = assuming that the not detected results are equal to 0 (ND = 0). Middle bound (MB) = assuming that the not detected results correspond to half the LOD (ND = 2.5 ng kg^−1^). Upper bound (UB) = assuming that the not detected results correspond to the LOD (ND = 5.0 ng kg^−1^). P95 = 95th percentile. M = male. F = female.

**Table 5 nutrients-13-04450-t005:** The margin of exposure (MOE) values based on dietary exposure to AFM1 for two age categories.

Food Group	MOE											
	Toddlers, 1–3 Years						Children, 3–9 Years					
	Male			Female			Male			Female		
	LB	UB	P95	LB	UB	P95	LB	UB	P95	LB	UB	P95
Infant formula	1,076,492	292,923	137,791	1,085,686	295,425	138,968						
Fermented milk products	43,819	30,958	7365	46,890	33,127	7881	65,393	46,200	10,991	66,566	47,028	11,188
Butter	2,206,810	1,183,032	244,158	1,886,792	1,011,476	208,752	2,322,206	1,244,894	256,925	2,301,937	1,234,028	254,682
Milk beverages	57,696	29,692	10,586	57,851	29,772	10,614	103,015	53,015	18,901	113,886	58,610	20,896
Sour cream	541,063	333,930	77,778	431,381	266,237	62,011	839,146	517,899	120,627	924,455	570,549	132,890
Cheese	1,830,065	396,320	497,778	1,567,020	339,355	426,230	2,596,110	562,215	706,142	2,679,887	580,358	728,929
Pasteurized and UHT milk	24,368	23,264	10,271	20,773	19,832	8756	43,166	41,211	18,195	45,920	43,840	19,356
Whey liquid							25,911	22,642	20,105			
**Average**	**825,759**	**327,160**	**140,818**	**728,056**	**285,032**	**123,316**	**856,421**	**355,439**	**164,555**	**1,022,109**	**422,402**	**194,657**

MOE calculations were based on benchmark dose (BMDL_10_) for AFB1 of 0.4 μg kg^−1^ bw day^−1^ and potency factor for AFM1 of 0.1 [30]. Lower bound (LB) = assuming that the not detected results are equal to 0 (ND = 0). Middle bound (MB) = assuming that the not detected results correspond to half the LOD (ND = 2.5 ng kg^−1^). Upper bound (UB) = assuming that the not detected results correspond to the LOD (ND = 5.0 ng kg^−1^). P95 = 95th percentile.

**Table 6 nutrients-13-04450-t006:** Cancer risk estimates calculated from chronic dietary exposure to AFM1. Scenario 1 (mean).

Food Group	Liver Cancer Risk (Case/100,000 Persons)											
	Toddlers, 1–3 Years						Children, 3–9 Years					
	Male			Female			Male			Female		
	LB	UB	P95	LB	UB	P95	LB	UB	P95	LB	UB	P95
Infant formula	0.00001	0.00003	0.00006	0.00001	0.00003	0.00006						
Fermented milk products	0.00018	0.00025	0.00106	0.00017	0.00024	0.00099	0.00012	0.00017	0.00071	0.00012	0.00017	0.00070
Butter	0.00000	0.00001	0.00003	0.00000	0.00001	0.00004	0.00000	0.00001	0.00003	0.00000	0.00001	0.00003
Milk beverages	0.00014	0.00026	0.00074	0.00013	0.00026	0.00074	0.00008	0.00015	0.00041	0.00007	0.00013	0.00037
Sour cream	0.00001	0.00002	0.00010	0.00002	0.00003	0.00013	0.00001	0.00002	0.00006	0.00001	0.00001	0.00006
Cheese	0.00000	0.00002	0.00002	0.00000	0.00002	0.00002	0.00000	0.00001	0.00001	0.00000	0.00001	0.00001
Pasteurized and UHT milk	0.00032	0.00034	0.00076	0.00038	0.00039	0.00089	0.00018	0.00019	0.00043	0.00017	0.00018	0.00040
Whey liquid							0.00030	0.00034	0.00039			
**Total**	**0.00066**	**0.00093**	**0.00276**	**0.00071**	**0.00098**	**0.00286**	**0.00069**	**0.00089**	**0.00166**	**0.00037**	**0.00037**	**0.00158**

Potency estimates of 0.0017 (mean) per 100,000 person-years per ng kg^−1^ bw day^−1^ were calculated for HBsAg-negative individuals. For HBsAg-positive individuals, potency estimates of 0.0269 (mean) per 100,000 person-years per ng kg^−1^ bw day^−1^ were calculated [31]. The risk of liver cancer was estimated as new cancer cases year^−1^ per 100,000 population by multiplying the AFM1 EDI by the average HCC potency 0.001952 (mean) based on 1% prevalence of HBV infection in Serbia.

**Table 7 nutrients-13-04450-t007:** Cancer risk estimates calculated from the chronic dietary exposure to AFM1. Scenario 2 (95% upper bound (UB)).

Food Group	Liver Cancer Risk (Case/100,000 Persons)											
	Toddlers, 1–3 Years						Children, 3–9 Years					
	Male			Female			Male			Female		
	LB	UB	P95	LB	UB	P95	LB	UB	P95	LB	UB	P95
Infant formula	0.00002	0.00007	0.00016	0.00002	0.00007	0.00016						
Fermented milk products	0.00049	0.00070	0.00294	0.00046	0.00065	0.00275	0.00033	0.00047	0.00197	0.00033	0.00046	0.00194
Butter	0.00001	0.00002	0.00009	0.00001	0.00002	0.00010	0.00001	0.00002	0.00008	0.00001	0.00002	0.00009
Milk beverages	0.00038	0.00073	0.00205	0.00037	0.00073	0.00204	0.00021	0.00041	0.00115	0.00019	0.00037	0.00104
Sour cream	0.00004	0.00006	0.00028	0.00005	0.00008	0.00035	0.00003	0.00004	0.00018	0.00002	0.00004	0.00016
Cheese	0.00001	0.00005	0.00004	0.00001	0.00006	0.00005	0.00001	0.00004	0.00003	0.00001	0.00004	0.00003
Pasteurized and UHT milk	0.00089	0.00093	0.00211	0.00104	0.00109	0.00247	0.00050	0.00053	0.00119	0.00047	0.00049	0.00112
Whey liquid							0.00084	0.00096	0.00108			
**Total**	**0.00184**	**0.00257**	**0.00766**	**0.00197**	**0.00271**	**0.00792**	**0.00109**	**0.00150**	**0.00460**	**0.00103**	**0.00142**	**0.00437**

Potency estimates of 0.0049 (95% upper bound (UB)) per 100,000 person-years per ng kg^−1^ bw day^−1^ were calculated for HBsAg-negative individuals. For HBsAg-positive individuals, potency estimates of 0.0562 (95% UB) per 100,000 person-years per ng kg^−1^ bw day^−1^ were calculated [31]. The risk of liver cancer was estimated as new cancer cases year^−1^ per 100,000 population by multiplying the AFM1 EDI by the average HCC potency 0.005413 (UB) based on 1% prevalence of HBV infection in Serbia.

## Data Availability

Results attained in this study are included in the manuscript. Individual data are not publicly available due to ethical restrictions.

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
