# Peer review of "Dietary Exposure and Risk Assessment of Aflatoxin M1 for Children Aged 1 to 9 Years Old in Serbia"

_nutrients, 2021, doi:10.3390/nu13124450_

Round 1

Reviewer 1 Report

Current research evaluated AFM1 from milk and milk-based food for Serbia children.  The research topic provides information to the scientific community; However, I have several comments:

  1. Current research has a total of 3404 food samples with 10 types. the sample size of each subtype is diverse, leading a result that AFM1 detection rate from lowest 2% to highest 79%, which is hard to compare the Aflatoxin M1 incidence between groups and further analysis.
  2. Due to the limited number, this study may lack representativeness of Serbia, it's better to treat the current paper as a report.
  3. The current study through the MOE approach and population risk to children population and found a conclusion that AFM1 exposure and health risk through milk products, could be negligible. This conclusion I think should be tested and verified by a bigger sample size research.

Author Response

Dear editors, first of all thank you for taking the paper into consideration. We also thank the reviewer for constructive advice. In accordance with the reviewer suggestions, we provide you with the answers point by point. We hope that the answers will satisfy your queries.

We are looking to hear your response asap.

Sincerely

Dragan Milicevic

Current research evaluated AFM1 from milk and milk-based food for Serbia children.  The research topic provides information to the scientific community; However, I have several comments:

  1. Current research has a total of 3404 food samples with 10 types. the sample size of each subtype is diverse, leading a result that AFM1 detection rate from lowest 2% to highest 79%, which is hard to compare the Aflatoxin M1 incidence between groups and further analysis.

Answer: Research has been carry-out during three years’ period of investigation. During that period, samples of raw milk, processed milk and dairy products were received in the laboratory of institute for analysis. Among them, raw milk was majority of the all samples. For risk assessment purposes we taken only processed milk, considering their intention. Raw milk did not take into the consideration. According to Serbian regulation, other milk products cheese, etc. are not mandatory to aflatoxin analysis. Thus, some group of milk products were in small number.

  1. Due to the limited number, this study may lack representativeness of Serbia, it's better to treat the current paper as a report.

Answer: this question is for editor.

  1. The current study through the MOE approach and population risk to children population and found a conclusion that AFM1 exposure and health risk through milk products, could be negligible. This conclusion I think should be tested and verified by a bigger sample size research.

Answer: Answer on this question is partly provided above. The reason for neglect risk lies in the fact that level of contamination is low, as well as level of consumption of milk by children. In addition, national food consumption survey was carry out according to EFSA protocol. Calculation confirm such hypothesis. Moreover, continuous monitoring is required.

Reviewer 2 Report

Below I present my comments:

Please, correct these points: 

a) The diagrams in Figure 1 should be corrected for better readability.

b) The ‘values of EDI’ abbreviation should be expanded (Line 20, Abstract).

c) The ‘MRL’ abbreviation should be expanded (Line 143).

Please answer the following questions: 

a) What are the possibilities to reduce the concentration of aflatoxins in products and thus improve their quality?

b) Can the conclusions described by the authors also apply to children from other European countries (not only from Serbia)?

c) Is it possible to test the level of accumulation of aflatoxins in the body?

Author Response

Dear editors, first of all thank you for taking the paper into consideration. We also thank the reviewer for constructive advice. In accordance with the reviewer suggestions, we provide you with the answers point by point. We hope that the answers will satisfy your queries.

We are looking to hear your response asap.

Sincerely

Dragan Milicevic

  1. a) The diagrams in Figure 1 should be corrected for better readability. Corrected accordingly.
  2. b) The ‘values of EDI’ abbreviation should be expanded (Line 20, Abstract). Corrected accordingly.
  3. c) The ‘MRL’ abbreviation should be expanded (Line 143). Corrected accordingly.

Please answer the following questions:

  1. What are the possibilities to reduce the concentration of aflatoxins in products and thus improve their quality?

Answer: Pre harvest, preventive measures and storage condition (humidity, temperature and ventilation) are the main factors to reducing mycotoxins contamination. Also, implementation of food safety control system such as HACCP are mandatory to prevent enter AFs along to the food chain.

  1. Can the conclusions described by the authors also apply to children from other European countries (not only from Serbia)?

Answer: derived conclusion are not of local interest, because particular situation could be changed depending of the year, season, climate condition etc.

  1. Is it possible to test the level of accumulation of aflatoxins in the body?

Answer: This approach more precisely reflects ingested amount of aflatoxins, but not only from milk, also from others sources such as vegetables. However, contribution of consumed foodstuffs to aflatoxins exposure is necessary. 

Round 2

Reviewer 1 Report

Introduction
1- Very lengthy introduction with may redundant parts.
2- The introduction lacks a clear research hypothesis or question.
Methods
3- Lines 170-171: All details related to the analytical methods, validation, and quality assurance have been previously described by Milievic et al. [5,25,26].
This needs to write some details of the methods.
4- Lines 310-312: A Shapiro–Wilk test of normality was run to check the normality of data and after recording the data as normal, a further test was used for statistical evaluation of the data. 
What exactly were these statistical tests?
Discussion
The manuscript lacks the Limitations and Strengths section. 

Author Response

Dear, following your request regarding to revision of manuscript, manuscript is revised according to reviewer suggestion as follows. We believe that in such form is ready for publication.

Sincerely

Dragan Milićević

Comments and Suggestions for Authors

Introduction1- Very lengthy introduction with may redundant parts.

Answer. Some parts in section Introduction which were repeatable in following part of manuscript were omitted.

2- The introduction lacks a clear research hypothesis or question. Answer.We believe that in present form looking better.

Methods

3- Lines 170-171: All details related to the analytical methods, validation, and quality assurance have been previously described by Milievic et al. [5,25,26]. This needs to write some details of the methods. Answer. The method is described in detail.

4- Lines 310-312: A Shapiro–Wilk test of normality was run to check the normality of data and after recording the data as normal, a further test was used for statistical evaluation of the data. 

Answer: For statistical analysis first step is to using a normality test to determine whether variable is normally distributed in some population. The Shapiro-Wilk test examines if a variable is normally distributed in some population. For statistical analysis first step is to using a normality test to determine whether variable is normally distributed in some population. The Shapiro-Wilk test examines if a variable is normally distributed in some population (frequency distribution).

In our study it is important supporting tool for right presentation and interpretation of obtained results, particularly for further risk assessment. Even more, Normality test is prerequisite activities for properly using mean or median, Q1, Q3, LB, MB or UB values. Examples are shown in figure 1 and 2.

What exactly were these statistical tests?

Discussion

The manuscript lacks the Limitations and Strengths section. Answer: this section was added.
